# Pb-resistant *Pantoea rwandensis* promotes maize's growth by altering Pb accumulation in biomass and soil Pb immobilization

**Runhai Jiang, Chengqiang Zhu, Shaofu Wen, Mei Zhang, Yani Zhang, Xiuli Hou** *

Yunnan Collaborative Innovation Center for Plateau Lake Ecology and Environmental Health, College of Agronomy and Life Sciences, Kunming University, Kunming, Yunnan, China

* hxlyn@aliyun.com

**Data Availability Statement:** All relevant data are within the paper and its Supporting Information files.

## Abstract

In lead (Pb)-contaminated soil, inoculating phosphate-solubilizing bacteria (PSB) to reduce the phytoavailability of Pb and to change the soil nutrients is an important way to inhibit the Pb uptake by plants. In this study, we isolated the native Pb-resistant *Pantoea rwandensis* from a tailings site. We employed broth culture and pot experiments to investigate the effect of the inoculation of *P. rwandensis* on Pb immobilization in the soil, the soil nutrients, the microbial community, and Pb accumulation in the biomass of maize (*Zea mays* L.). The results showed that *P. rwandensis* not only tolerated $Pb^{2+}$ but also dissolved $Ca_3(PO_4)_2$ by secreting oxalic acid, tartaric acid, lactic acid, and acetic acid, which significantly increased the amount of dissolved P, up to 567.50 $mg \cdot L^{-1}$. In the pot experiment, the application of culture filtrate, an inoculation with *P. rwandensis*, or a mixture of culture filtrate and *P. rwandensis* increased the soil residual Pb by 49.57%, 89.81%, and 41.69%, respectively, compared to the control. Consequently, the increase in the residual Pb in the soil inhibited its uptake by maize, and an inoculation of *P. rwandensis* or a mixture of culture filtrate and *P. rwandensis* decreased the accumulation of Pb in the shoot biomass of maize by 61.65% and 72.48% and in the roots of maize by 26.00% and 39.59%, respectively. Meanwhile, *P. rwandensis* and the mixture of culture filtrate and *P. rwandensis* increased the shoot biomass of maize by 54.39% and 17.86% and the root biomass of maize by 108.77% and 17.86%, respectively. A *P. rwandensis* inoculation increased the biodiversity of the fungal community and the abundance of *Proteobacteria*, *Mortierella*, and *Sphingomonas*. Our results demonstrate that an inoculation with *P. rwandensis* could promote the transformation of Pb fractions and a functional change in the soil bacterial community, ultimately decreasing Pb accumulation in the biomass and promoting the development of maize.

## Introduction

Soil contamination with lead (Pb) is a global environmental issue because of mining and smelting [1, 2]. Its toxicity and enrichment in animals and plants has generated wide attention. Chemical precipitation, absorption, and ion exchange to immobilize Pb in soil are effective

**Funding:** This work is funded by National Natural Science Foundation of China (42167009), Joint Fund Project of Yunnan Provincial Universities (2018FH001-004), and Scientific Research Fund Project of Yunnan Provincial Department of Education (2022Y712, 2023Y0874).

**Competing interests:** The authors have declared that no competing interests exist.

methods for decreasing the uptake and accumulation of Pb by plants [3–5]. According to the European Community Bureau of Reference sequential extraction method, heavy metals are divided into four fractions: a soluble/exchangeable fraction (F1), a reducible fraction (F2), an oxidizable fraction (F3), and a residual fraction (F4). F4 is recognized as the most stable fraction. Nowadays, several phosphate-based amendments, such as dicalcium phosphate, diammonium phosphate, mono-ammonium phosphate, phosphoric acid, apatite, and hydroxyapatite, are applied to promote the transformation of Pb from an unstable fraction (F1, F2, and F3) to a mineral structure (F4) for the immobilization remediation of Pb-contaminated soil [6, 7]. The phosphate-based amendments induce the formation of various Pb phosphorate compounds, such as $Pb_5(PO_4)_3X$ [X = F (fluorine), Cl (chlorine), Br (bromine), or OH (hydroxy)], in contaminated soils, thus reducing the Pb bioavailability. However, some factors, such as the solubility of the phosphorus (P) sources, the pH value, the Pb speciation, and the particle size, could interact and make Pb immobilization more difficult. To ensure a high solubility of P, an excessive dose of P compounds is used in the soil to ensure sufficient soluble orthophosphate anions for Pb immobilization [8]. However, an excessive amount of P compounds may result in the eutrophication of water environments [9–12]. Therefore, improving P solubility for Pb immobilization in Pb-contaminated soils in an environmentally friendly way is a topical issue.

Phosphate-solubilizing bacteria (PSB) are widely distributed in soil, driving phosphorus (P) transformations and dominating the composition of the P fraction. PSB can convert P, which is difficult for plants to take up, into bioavailable P. These bacteria not only enhance the utilization efficiency of P for plants and improve the disease resistance of plants, but they also improve the soil structure [13]. Inorganic P compounds are solubilized by PSB mainly through the production of organic acids, which reduce the pH of the rhizosphere soil or chelate the cations that are responsible for P precipitation [13–16]. In addition, PSB can enhance soil acid phosphatase and alkaline phosphatase through enzymatic hydrolysis to mineralize soil organic P [17, 18]. Numerous studies have demonstrated that PSB can enhance plant growth by elevating the levels of bioavailable P in soil, as well as by producing beneficial metabolites such as phytohormones and siderophores, particularly in soils with a low fertility [19–21]. For example, *Pantoea* possesses an astonishing number of plant-growth-promoting genes, including those involved in nitrogen fixation, phosphate solubilization, 1-aminocyclopropane-1-carboxylic acid deaminase activity, indoleacetic acid and cytokinin biosynthesis, and jasmonic acid metabolism [22]. Additionally, *Pantoea* possesses many functional plant-growth-promoting mechanisms and a heavy metal tolerance. It promotes nodulation processes in leguminous plants under stress and can also enhance the growth of other plants while suppressing plant pathogens. *Pantoea* is capable of altering the entire soil bacterial community, significantly affecting its structure and composition and particularly influencing the relative abundance of microbial communities such as those belonging to the Firmicutes phylum [23–25]. Apart from their functional roles in increasing the amount of soil-available P and promoting plant growth, the increase in soluble P in soil provides the conditions necessary for the formation of Pb phosphate compounds in Pb-contaminated soil due to the accelerated release of phosphate from native insoluble P species in the soil (mainly Fe–P). Currently, PSB are used to immobilize Pb through the mineralization of phosphate [26]. Pb immobilization in soil inhibits the Pb uptake by plants and decreases the toxicity of Pb [27–29].

Farmers have applied fertilizer to Pb-contaminated agricultural fields for decades, believing that sufficient nutrients would promote the growth of crops, especially in the Jinding Pb-Zn mining area, Yunnan Province, southwestern China, which is the richest reserve of Zn in Asia [30, 31]. In this area, large amounts of P applied as fertilizer have been transformed into various unavailable forms through precipitation with the highly reactive $Al^{3+}$ and $Fe^{3+}$ in the soil [32]. The farmland near the tailings has been exposed to heavy metal pollution for a long time,

resulting in particularly severe Pb contamination in the soil. Therefore, the use of inoculated Pb-resistant PSB in Pb-contaminated soil to facilitate P uptake by plants and the immobilization of Pb is an effective way for the local people to achieve sustainable development. Although extensive research has focused on plant–microbe interactions, these studies have only concentrated on the effects of bacterial isolates on promoting plant growth and soil properties. However, how the culture filtrate, the bacterial isolate, or the combined application of both specifically affects maize (*Zea mays* L.) growth, the soil properties, the Pb fraction, and the soil microbial diversity remains unclear. We screened an indigenous PSB, *Pantoea rwandensis*, from tailings with a remarkable ability to dissolve inorganic P. This work aimed to provide a deeper understanding of how this bacterium affects the growth of maize, which is extensively planted worldwide. By studying how *P. rwandensis* and its metabolic products optimize maize's stress tolerance response, we hoped to explore new methods to support sustainable agricultural practices. Therefore, in the present study, the mechanisms by which *P. rwandensis* and its metabolic products immobilize the heavy metal Pb, improve the soil quality, regulate the soil microbial community structure, and thereby promote maize growth were analyzed. The effectiveness of *P. rwandensis* and its metabolic products in remediating Pb-contaminated soils was assessed. This research enhances our understanding of how *P. rwandensis* immobilizes the heavy metal Pb to alleviate the stress on maize. The application of *P. rwandensis* can provide a foundation for the remediation of heavy-metal-contaminated soils and an improvement in maize quality.

## Materials and methods

### Study site and soil sampling

The soil samples for this study were collected from the Jinding Pb–Zn tailings site, located in Nujiang Lisu Autonomous Prefecture, Yunnan Province, southwestern China. The geographical location is N 26˚25' and E 99˚25', and the site has an altitude of 2380 m. The tailings site has a temperate climate with an average annual temperature of 10.4 to 11.8˚C and an average annual precipitation of 1088 mm. Its Pb-Zn reserves, with 15.4761 million tons of Pb and Zn, rank the fifth in the world. Due to the long-term mining, plants struggle to grow in the soil in this area [33]. Our study site did not involve endangered or protected species, so no specific permissions were required for the location/activity.

Healthy plants of *Cirsium monocephalum* growing on tailings were selected. We used a sterile trowel to excavate the plant samples with complete root systems, taking care not to damage the roots during excavation. Then, the plants were gently shaken to dislodge soil near the root system, and a sterile brush was used to remove the soil adhering to the roots and transfer it into a sterile bag. The soil samples were stored at 4˚C and quickly transported to the laboratory to screen for and isolate PSB.

### Isolation of bacteria

A sample of 2 grams of soil was suspended in 18 mL of sterile water. After shaking the sample for 30 minutes and letting it settle for 5 minutes, 250 µL of the liquid phase was inoculated on Pikovskaya (PVK) agar medium using the serial dilution counting method [34]. The bacterial cultures were incubated at 28˚C for a period of 4–7 days, during which clear halo zones formed around individual bacteria, indicating distinct single colonies without any merging. The bacteria exhibiting these clear zones were successfully isolated and transferred to Luria–Bertani (LB) medium for cultivation. To purify the PSB, the streak plate method was employed to obtain single colonies. Subsequently, these purified colonies were maintained on LB agar plates and stored at 4˚C. We picked up the PSB from the preserved LB medium using a sterile

inoculation loop, streaked it onto a PVK agar plate, and incubated the plate at 28˚C for 7 days. The diameter of the bacterial colony and the halo zone around it were measured, and the phosphate solubilization index (PSI) was used to qualitatively characterize the bacterium's phosphate solubilization ability. The PSI is calculated as the ratio of the diameter of the halo zone to that of the bacterial colony [35]. The calculation formula is as follows:

$PSI = D/d$ where

$d$—colony diameter (mm);

$D$—halo zone diameter (mm).

## Acid production and phosphate solubilization of bacteria

The screening PSB were activated and cultured in LB liquid medium at 180 r·min$^{-1}$ for 30 min (28˚C). The bacterial suspension was adjusted to a concentration of approximately $1 \times 10^8$ CFU·mL$^{-1}$ of bacteria using sterile water. Then, 1 mL of the bacterial suspension was inoculated into PVK broth medium containing 1 g of $Ca_3(PO_4)_2$ per 100 mL, which served as the sole P source. In the control treatment, all components were the same except that sterile water was used instead of the PSB suspension. The broth medium was continuously incubated at 180 r·min$^{-1}$ for 7 days (28˚C). A sample of the broth medium (20 mL) was taken every day from each flask, centrifuged, and filtered through a 0.45 mm filter membrane. The pH of the filtrate was measured, and its dissolved P content was quantified using the molybdenum antimony colorimetric method [36]. A Pearson correlation analysis was used to analyze the correlation coefficient between the dissolved P and the pH in the culture medium. The dissolution rate of $Ca_3(PO_4)_2$ was calculated using the following formula:

$$c(\%) = \frac{m}{M} \times 100$$

where

c: the dissolution rate of $Ca_3(PO_4)_2$ (%);

m: the content of dissolved P in the broth medium (mg·L$^{-1}$);

M: the content of $Ca_3(PO_4)_2$ in the broth medium (mg·L$^{-1}$).

On the 7th day, in order to analyze the component of organic acids, the cell-free supernatant was obtained and subjected to high-performance liquid chromatography (HPLC), after centrifugation at 11739 g for 20 min and filtration through a 0.45 μm membrane [37].

An amount of 1 mL of an overnight LB-medium culture of bacteria with a concentration of $1 \times 10^8$ CFU·mL$^{-1}$ was transferred into a 250 mL Erlenmeyer flask containing 80 mL of LB medium supplemented with sterile tryptophan. The liquid culture was grown at 28˚C under agitation (150 rpm), and the cells were separated from the exhausted medium using centrifugation (7513 g for 10 min) and discarded after 72 h. The collected supernatant was filtered through a 0.22 μm membrane. The total IAA was determined using a colorimetric method with the Salkowski reagent, and authentic IAA was used as a standard [25].

## PSB's tolerance to Pb

Firstly, 1 mL of a bacterial suspension containing $1 \times 10^8$ CFU·mL$^{-1}$ was added to PVK broth medium. In the control treatment, all the components in the medium were the same except that sterile water was used instead of the PSB suspension. The concentration of Pb$^{2+}$ (Pb $(NO_3)_2$) in the broth medium was 0, 250, 500, 750, or 1000 mg·L$^{-1}$. Using the control treatment as a reference, the bacterial liquid's OD$_{600}$ value was determined at 600 nm using an ultraviolet–visible spectrophotometer (DR 5000, USA) following seven days of culture at 180 r·min$^{-1}$ (28˚C).

## Microbiological identification of bacteria

The extract's genomic DNA of J101 was obtained using a total bacterial DNA extraction reagent. The primers 27F (5'-AGTTTGATCMTGGCTCAG-3') and 1492R (5'-GGTTACC TTGTTACGACTT-3') were used for the PCR amplification of the 16S rRNA gene [38]. The amplified products were quantified using 1% agarose gel electrophoresis and sequenced by Sangon Biotech Co., Ltd (Shanghai, CHN). The sequences were aligned to the GenBank database using the NCBI basic local alignment search tool (BLAST) (https://blast.ncbi.nlm.nih.gov/Blast.cgi). A phylogenetic tree was constructed using MEGA software (version 11; Mega, Raynham, MA, USA).

## Maize growth promotion experiment

**Soil sample.** The soil for the pot experiment was collected from farmland located near the Jinding Pb-Zn tailings site (N 26°27'37, E 99°28'32). Five soil samples were collected in September 2022 to a depth of 0–20 cm using a five-point sampling method. These soil samples were thoroughly mixed in a sterile polyethylene bag and then stored at room temperature away from light. The portion of the samples used to determine the background values of the soil was air-dried.

**Preparation of inoculation.** After the bacteria were cultured in the LB liquid medium for 72 h (28°C), sterile water was used to dilute a portion of the fermentation liquid to a viable cell density of $1\times10^8$ CFU·mL$^{-1}$. The other portion of the fermentation liquid was centrifuged at 4°C and 2935 g for 10 min, and the culture filtrate was collected. The bacterial cells were washed 3 times with sterile physiological saline (0.9% sodium chloride) and then adjusted to a viable cell density of $1\times108$ CFU·mL$^{-1}$ using sterile water. The fermentation liquid was a mixture of both the bacteria themselves and their secretions. The filtrate contained only the secretions from the bacteria, while the bacterial solution contained only the bacteria.

**Pot experimental design.** Maize seeds of the "Sidatian 221" variety were acquired from Beijing Zhongnong Star Agricultural Technology Development Co., Ltd. Initially, the seeds were immersed in a 95% ethanol solution for 30 seconds. This step was followed by a subsequent soaking in a 1% sodium hypochlorite solution for 10 minutes. Finally, the seeds were thoroughly rinsed multiple times with sterile distilled water to remove any residual sterilizing agents. The maize seeds were germinated, and then two seedlings were transplanted into flowerpots containing 2.5 kg of soil in a greenhouse with a light intensity of 4048 Lux, daytime/nighttime temperatures of 30°C/25°C, and a humidity of 69.5% (the pot soil background values were as follows: total phosphorus (TP), 860.33±2.52 mg·kg-1; available phosphorus (AP), 12.07±3.70 mg·kg-1; alkaline hydrolysis nitrogen (AN), 65.33±4.04 mg·kg-1; Pb content, 1985.47 mg·kg-1; and pH, 7.61±0.0). A total of 4 treatments were set up in this experiment, with 6 replicates of each treatment: (1) maize watered with sterile water (control); (2) maize inoculated with the culture filtrate of the bacteria (J101CS); (3) maize inoculated with the bacteria (J101BS); and (4) maize inoculated with a mixture of both culture filtrate and bacteria (J101FL). Each replicate received 300 mL of inoculum. Watering occurred every 3 days, and the experiment lasted for 45 days.

## Plant and soil chemical analysis

**Chemical and microbial diversity analyses of soil samples.** We randomly selected three replicate samples from each treatment group for soil and plant sample analyses. A 20 g soil sample was collected from 1–2 centimeters near the maize root system, preserved in a sterile bag, and stored at 4°C with the purpose of analyzing the soil microbial community structure. The sequencing work was carried out by Shanghai Sangon Company using the Illumina MiSeq system (Illumina Inc., USA) according to the manufacturer's instructions. The soil samples

were dispersed and passed through a 0.149 mm sieve before measuring the pH; the concentrations of total phosphorus (TP), available phosphorus (AP), and alkaline hydrolysis nitrogen (AN); and the activity of urease (URE) and acid phosphatase (ACP), as described in [39]. The soil TP and AP were quantified using the molybdenum antimony colorimetric method, AN was determined using the alkali N-proliferation method, and URE was determined using the sodium phenol–sodium hypochlorite colorimetric method, whereas ACP was determined using disodium phenyl phosphate colorimetry. To extract heavy metals, air-dried soil samples were passed through a 0.149 mm nylon sieve and digested in 1:2:2 (V:V:V) $HNO_3$:$HCl$:$HClO_4$. Pb was measured using inductively coupled plasma mass spectroscopy (ICP-MS) [40]. The different Pb fractions in the soil were determined by extracting solutions according to the China Geological Survey's seven-step continuous extraction method (D2005-03, (cgs.gov.cn)).

**Chemical analyses of plant samples.**   The maize plants were harvested after 45 days; their shoot parts and roots were separated and rinsed thoroughly with deionized water. The length and fresh biomass of both parts were recorded, and the roots were scanned using a root scanner (Microtek Scanmaker i800 plus, CHN). The content of malondialdehyde (MDA), superoxide dismutase (SOD), and peroxidase (POD) activity of the maize leaves and roots were measured using reagent kits provided by the Nanjing Institute of Biotechnology Engineering. For the dry weight measurements, the maize samples were dried in an oven at 105˚C for 30 min until reaching a constant weight at 85˚C. Then, these dried plant samples were crushed into a fine powder with a particle size of less than 0.149 mm to determine the contents of nitrogen (N) and P according to the analysis method provided by Lu [39] and the Pb content according to the method described by Ozen and Yaman [41].

## Data analysis

The statistical analyses in this study were performed using SPSS 27.0. A one-way ANOVA (analysis of variance) was conducted to compare the means derived from the different treatments. The significance was calculated using Duncan's multiple range test at the $p < 0.05$ level. The 16S rRNA sequences were compared using the BLAST program through the National Center for Biotechnology Information (nih.gov), and then a phylogenetic tree was constructed using Molecular Evolutionary Genetics Analysis (MEGA, 11.0) software. A phylogenetic tree, also known as an evolutionary tree, refers to a diagram that represents the evolutionary relationships among various species believed to have a common ancestor. In a phylogenetic tree, if two species share a more recent common ancestor, their relationship is considered closer; conversely, if their common ancestor is more distant, the relatedness between the two species is considered lower. An analysis of the correlation among the antioxidant enzyme activity, N, P, and Pb accumulation in maize leaves and roots, as well as various indicators of maize biomass, was conducted using Canoco 5.0 software. Meanwhile, the soil microbial classification data were processed and examined through an online cloud platform available at ngs.sangon.com.

## Results

### P solubilization and Pb tolerance of PSB

In this study, four PSB were isolated from the rhizosphere soil of a *Cirsium monocephalum* plant. By comparing the PSI values of the four bacteria, one bacterium was found to exhibit a PSI value of 1.82, which was the highest among them. This bacterium was designated as J101 and subjected to further analyses. An inoculation with J101 increased the amount of dissolved P in the broth medium with incubation time (Fig 1A). The highest amount of dissolved P was observed in the broth medium after 4 days of inoculation with J101, with a value of 567.50 mg·L$^{-1}$, resulting in a dissolution rate for $Ca_3(PO_4)_2$ of 56.75%. However, after 5 days, the

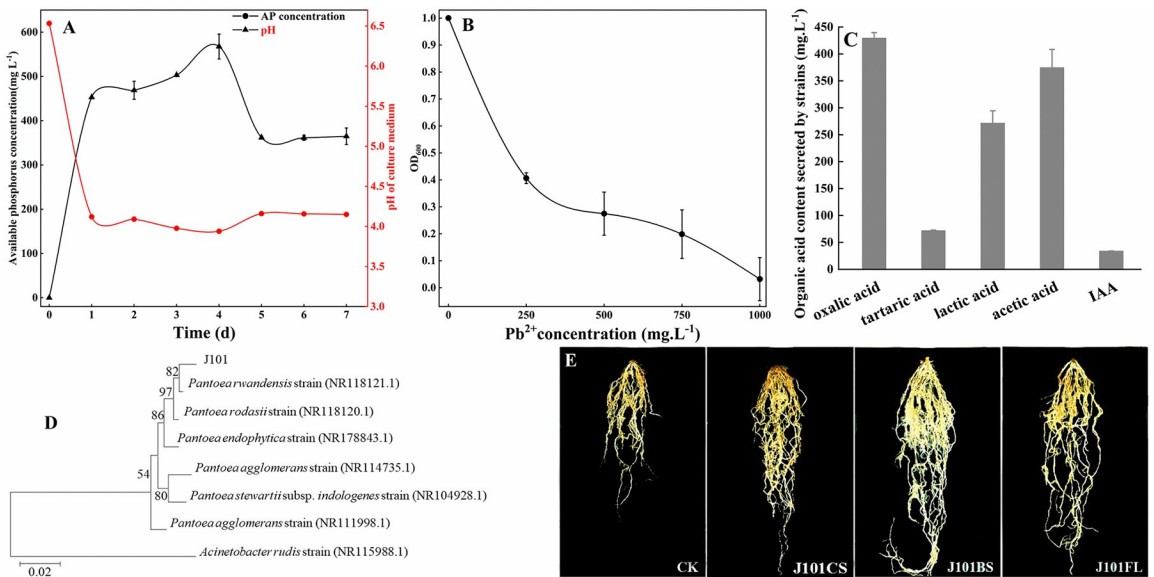

**Fig 1. Analysis of dissolved P, Pb tolerance, and secreted metabolites of J101.** (A) The change in the dissolved P content and the pH value in the broth medium; (B) the OD values at 600 nm after culturing the PSB in fermentation liquid with $Pb^{2+}$ concentrations of 0, 250, 500, 750, or 1000 mg·L$^{-1}$ for 72 h; (C) the content of organic acid and IAA secreted by the PSB; (D) the phylogenetic tree of J101 based on the 16S rDNA gene sequences; and (E) the root systems of maize under different treatments.

amount of dissolved P showed a decreasing trend. Meanwhile, there was a significant decrease in the pH of the broth medium over time, with the lowest pH value being 3.94 after 4 days. A Pearson correlation analysis revealed a highly significant negative correlation between the amount of dissolved P and the pH (r = -0.925**, $p < 0.01$).

When the PSB J101 was grown in broth media with different concentrations (0, 250, 500, 750, or 1000 mg·L$^{-1}$) of $Pb^{2+}$, we found that the growth of PSB J101 was inhibited by the highest $Pb^{2+}$ concentration (Fig 1B). However, at a concentration of 250 mg·L$^{-1}$ of $Pb^{2+}$, the PSB J101 showed 40.6% of its activity without Pb in the broth medium. When this bacterium was exposed to a concentration of 750 mg·L$^{-1}$ of $Pb^{2+}$, it maintained its growth, although its OD$_{600}$ values decreased, indicating that J101 has a certain tolerance to $Pb^{2+}$.

The organic acids secreted by PSB were responsible for the dissolved P from $Ca_3(PO_4)_2$ (Fig 1C). When $Ca_3(PO_4)_2$ was the sole source of P in the broth medium, the PSB J101 secreted oxalic acid, tartaric acid, lactic acid, and acetic acid. Moreover, the bacteria J101 can also secrete IAA (Fig 1C).

## The l6S rDNA gene of PSB

A 1478 bp 16S rRNA gene fragment was obtained from J101. A phylogenetic tree was constructed based on a comparison of the alignment results with the data in GenBank. The phylogenetic tree (shown in Fig 1D) summarizing the phylogenetic relationship among *Pantoea* species revealed that J101 is closely related to *Pantoea rwandensis*, with a 98.81% similarity. Considering its morphological, physiological, and biochemical characteristics, and a phylogenetic analysis based on the 16S rDNA gene sequence, J101 should be classified into *P. rwandensis*.

## Promotion of maize growth by PSB

In the three inoculation treatments, the application of inoculated culture filtrate (J101CS), bacterial isolates (J101BS), or a combination of culture filtrate and bacterial isolates (J101FL)

**Table 1. Effects of different treatments on agronomic traits and biomass of maize.**

|  | Plant height (cm) | Stem thickness (mm) | Fresh weight of shoot (g) | Fresh weight of roots (g) | Dry weight of shoot (g) | Dry weight of roots (g) |
|---|---|---|---|---|---|---|
| CK | 26.30±1.64a | 5.29±0.75a | 3.28±0.84a | 2.21±0.59a | 0.57±0.14a | 0.56±0.07a |
| J101CS | 31.25±7.04b | 6.86±1.35b | 5.56±1.36b | 6.08±1.22b | 0.85±0.31b | 0.47±0.05a |
| J101BS | 31.83±7.34b | 6.28±0.72b | 5.78±0.90b | 6.52±0.98b | 0.88±0.08b | 0.66±0.07b |
| J101FL | 31.91±2.55b | 7.00±1.04b | 7.59±1.97c | 9.16±2.79c | 1.19±0.38c | 0.66±0.12b |

**Note:** Different lowercase letters in the table column-wise indicate significant differences between groups at $p < 0.05$.

significantly promoted maize growth. This was evidenced by increases in the plant height, stem thickness, fresh weight, and dry weight of maize (Table 1), as well as a notable improvement in root biomass. In summary, under these experimental conditions, the use of bacteria or their culture filtrate can effectively enhance multiple growth parameters in maize (Fig 1E).

## The enrichment of Pb and the change in antioxidant enzyme activity in maize

The application of the culture filtrate (J101CS), inoculation with bacteria (J101BS), or treatment with a mixture of culture filtrate and bacteria (J101FL) reduced the accumulation of Pb in maize leaves by 69.40%, 61.65%, and 72.48%, respectively, compared to the un-inoculated treatment group (Fig 2A). An inoculation with J101BS or J101FL also significantly decreased the accumulation of Pb in maize roots by 26.00% and 39.59%, respectively. The SOD activity in maize leaves decreased by 11.79% in the J101CS treatment compared to the un-inoculated treatment group. An inoculation with bacteria (J101BS) or a mixture of culture filtrate and bacteria (J101FL) increased the SOD activity in maize roots by 9.46%, 3.71%, and 9.86%, respectively (Fig 2B) and increased the POD activity in maize roots by 4.58%, 6.25%, and 2.27%, respectively, compared to the un-inoculated treatment group (Fig 2C). In the J101BS and J101FL treatment groups, the MDA content in the maize leaves was significantly lower than in the control group. The MDA content of maize roots showed a significant decrease in the J101CS, J101BS, and J101FL treatment groups; in particular, the lowest MDA content was observed in the J101FL treatment group, with a value of 0.45 nmol·g·prot$^{-1}$, which was 95.66% lower than that of the un-inoculated treatment group (Fig 2D).

## Nutrient uptake by maize

In the J101CS and J101FL treatment groups, the N content in the maize shoots and roots significantly increased compared to that of the un-inoculated treatment group (Fig 3A), and the N content in the shoots and roots increased by 88.89% and 37.04%, and by 80.95% and 35.71%, respectively. The P content in the maize shoots was not significantly different among the three treatment groups. However, in the J101CS treatment group, the P content in the maize roots increased by 23.58% compared to the CK group, although the difference was not significant (Fig 3B).

## Soil properties

Heavy metals are divided into seven fractions in soil: water-soluble, ion-exchange, carbonate-bound, humic-acid-bound, iron-manganese-oxide-bound, strong organic-bound, and residual forms, and the mobility of these seven fractions of Pb decreases in the following order: water-soluble>ion-exchange>carbonate-bound>humic-acid-bound>iron-manganese-oxide-bound>strong organic-bound>residual forms. We found that the application of culture

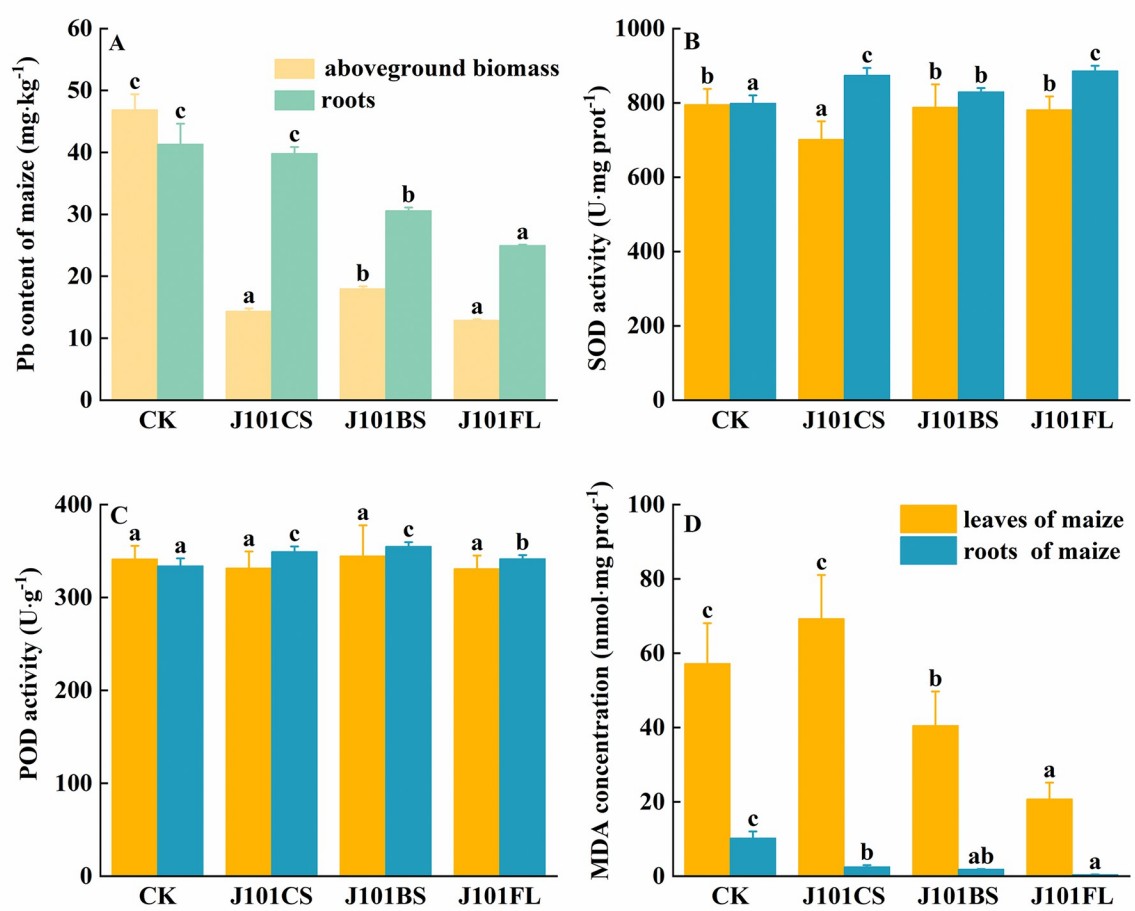

**Fig 2. The effects of different treatments on Pb uptake, antioxidant enzyme activity, and malondialdehyde content in maize.** (A) Pb absorption by maize; (B) changes in superoxide dismutase (SOD) activity in maize; (C) changes in peroxidase (POD) activity in maize; and (D) changes in malondialdehyde (MDA) content in maize. **Note:** Different lowercase letters indicate significant differences between groups.

filtrate (J101CS), an inoculation with bacteria (J101BS), and the mixture of culture filtrate and bacteria (J101FL) changed the soil Pb fractions. In these three treatment groups, the content of ion-exchange Pb in the soil was reduced by 14.08%, 14.41%, and 13.56%, respectively (Table 2). However, the fraction of carbonate-bound, humic-acid-bound, iron-manganese-oxide-bound, and strong organic-bound Pb in the soil was significantly increased. In the J101CS, J101BS, and J101FL treatment groups, the carbonate-bound Pb content was increased by 62.50%, 35.06%, and 24.17%; the content of humic-acid-bound Pb was increased by 14.11%, 9.95%, and 8.06%; the iron-manganese-oxide-bound Pb was increased by 12.59%, 8.04%, and 19.93%; and the strong organic-bound Pb increased by 39.73%, 44.01%, and 18.17%, respectively. The fraction of residual Pb content in the soil increased significantly by 49.57%, 89.81%, and 41.69% in the J101CS, J101BS, and J101FL treatment groups. These results indicate that PSB helped transform the mobilization-fraction Pb into immobilization-fraction Pb.

In the pot experiment, we found that the application of culture filtrate (J101CS) or an inoculation with bacteria (J101BS) significantly increased the activity of the acid phosphatase (ACP) activity in the soil by 35.19% and 42.59%, respectively, compared to the un-inoculated

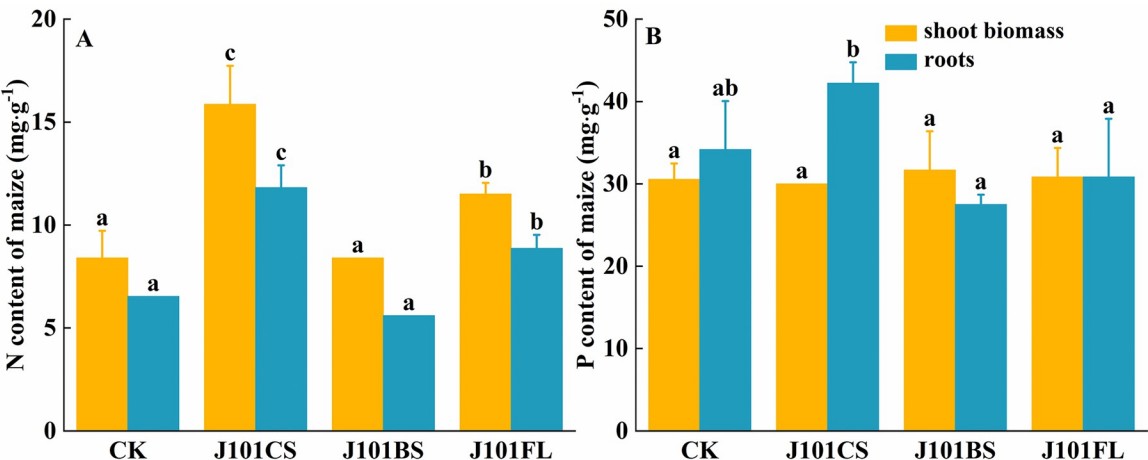

**Fig 3. Changes in the uptake of the nutrient elements N and P by maize.** (A) N uptake by maize; (B) P uptake by maize. Note: Different lowercase letters indicate significant differences between groups.

treatment group. The application of culture filtrate (J101CS), the inoculation with bacteria (J101BS), and the mixture of culture filtrate and bacteria (J101FL) also significantly increased the content of available phosphorus (AP) in the soil by 44.93%, 37.92%, and 25.35%, respectively (Table 3). The application of culture filtrate did not significantly affect the pH of the soil. However, the inoculation of bacteria (J101BS) and the mixture of culture filtrate and bacteria (J101FL) significantly decreased the pH values of the soil compared to the un-inoculated treatment group. The application of culture filtrate (J101CS), the inoculation with bacteria (J101BS), and the mixture of culture filtrate and bacteria (J101FL) did not significantly affect the urease (URE) activity or the alkaline nitrogen in the soil.

## Rhizosphere soil microbial community diversity and structure

PSB can change the diversity and composition of soil microbial communities. In Pb-contaminated soils, there were no significant differences in the Chao and Simpson indices of the bacterial communities in the maize rhizosphere soil for the J101CS, J101BS, or J101FL treatment groups compared to the un-inoculated treatment (Table 4). However, the Chao index of fungal communities was significantly increased in the J101CS and J101FL treatment groups, while the Simpson index significantly decreased after the application of culture filtrate (J101CS) or an inoculation with bacteria (J101BS) (Table 4), indicating that the application of microbial

**Table 2. Effects of different treatments on Pb forms in soil.**

| The forms of heavy metal Pb (mg·kg$^{-1}$) | CK | J101CS | J101BS | J101FL |
|---|---|---|---|---|
| Water-soluble forms | 3.32±0.01a | 3.45±0.00c | 3.39±0.00b | 3.37±0.02b |
| Ion-exchange forms | 168.41±2.05b | 144.69±1.47a | 144.14±0.63a | 145.58±0.89a |
| Carbonate-bound forms | 431.36±1.88a | 700.98±11.91d | 582.58±13.41c | 535.61±24.37b |
| Humic acid-bound forms | 7.44±0.09a | 8.49±0.25b | 8.18±0.18b | 8.04±0.33b |
| Iron-manganese oxide-bound forms | 1064.62±45.63a | 1198.70±15.83b | 1150.18±25.75b | 1276.75±26.08c |
| Strong organic-bound forms | 240.12±2.57a | 335.52±21.88c | 345.80±17.49c | 283.76±22.11b |
| Residual forms | 70.19±3.09a | 104.98±0.38c | 133.23±1.70d | 99.45±2.43b |

**Note:** Different lowercase letters in the table row-wise indicate significant differences between groups at $p < 0.05$.

**Table 3. Effects of different inoculation on soil properties.**

| | URE activity (mg. g$^{-1}$, 24$^{-1}$) | ACP activity (mg. g$^{-1}$, 24h$^{-1}$) | AP (mg.kg$^{-1}$) | AN (mg.kg$^{-1}$) | pH value |
|---|---|---|---|---|---|
| CK | 18354.70±6379.09a | 270.00±33.00a | 14.40±1.68a | 133.00±9.90a | 7.15± 0.01b |
| J101CS | 15901.97±4259.94a | 365.00±30.64bc | 20.87 ±0.52c | 130.67±38.55a | 7.11± 0.04b |
| J101BS | 14519.37 ±3988.02a | 385.00 ±2.36c | 19.86 ±0.25c | 105.00±0.00a | 6.91± 0.05a |
| J101FL | 15030.87 ±4613.85a | 290.00±42.43±ab | 18.05 ±0.40b | 149.33±29.14a | 6.90± 0.06a |

**Note:** Different lowercase letters in the table column-wise indicate significant differences between groups at $p< 0.05$. AP: available phosphorus; AN: alkaline nitrogen; ACP: acid phosphatase activity; and URE: urease activity.

inoculants can significantly enhance the diversity of fungal communities in maize rhizosphere soil.

Bacteria from 10 phyla and fungi from 6 phyla were annotated, and the abundance of the top ten types at the phylum level and the genus level was analyzed (Fig 4). In maize rhizosphere soil, *Proteobacteria* was the dominant phylum. The relative abundance of the *Proteobacteria* phylum in the maize rhizosphere soil of the J101CS and J101FL treatment groups was significantly higher than that of the un-inoculated treatment group, followed by the relative abundance of *Acidobacteria* (Fig 4A). At the genus level, the J101CS and J101FL treatments increased the abundance of the dominant genus *sphingomonas* and the subdominant genus *gemmatimonas* in the bacterial community compared to the un-inoculated treatment group (Fig 4B). In the fungal communities of maize rhizosphere soil, *Ascomycota* was identified as the dominant phylum, with a relative abundance range of 59.52% to 71.48%. The relative abundance of *Ascomycota* in the J101BS and J101FL treatment groups was lower than that in the un-inoculated treatment group. However, there was a significant decrease in *Mortierellomycota*'s relative abundance in the J101FL treatment group (Fig 4C). At the genus level, *Oedocephalum* was identified as the dominant genus in maize rhizosphere soil, with a relative abundance range of 8.27% to 31.91%, followed by *Mortierella*, with a relative abundance range of 12.05% to 23.73% (Fig 4D). Compared to the un-inoculated treatment group, the application of culture filtrate (J101CS) increased the relative abundance of *Mortierella* in maize rhizosphere soil. However, in the J101CS, J101BS, and J101FL treatment groups, the relative abundance of *Oedocephalum* was increased.

## Linkages between maize growth and soil properties

To determine the relationship between soil properties and maize growth, we first used a detrended correspondence analysis (DCA) to identify the growth indicators of maize and the soil factors, and the maximum gradient length of the two sort axes was 0.24, which is less than

**Table 4. The α-diversity index of bacterial and fungal communities in maize rhizosphere soil.**

| Inoculation | Bacterial community | | Fungal community | |
|---|---|---|---|---|
| | Chao | Simpson | Chao | Simpson |
| CK | 2,765.35± 9.07a | 0.0086±0.0024a | 647.85±51.30 a | 0.27±0.17a |
| J101CS | 2,685.36± 65.86a | 0.0112±0.0047a | 768.77±17.67b | 0.05± 0.01b |
| J101BS | 2,720.58± 10.36a | 0.0085±0.0004a | 743.18±19.82ab | 0.05± 0.01b |
| J101FL | 2,723.07± 39.56a | 0.0077±0.0025a | 804.30±44.27b | 0.17± 0.03a |

**Note:** Different lowercase letters in the table column-wise indicate significant differences between groups at $p< 0.05$.

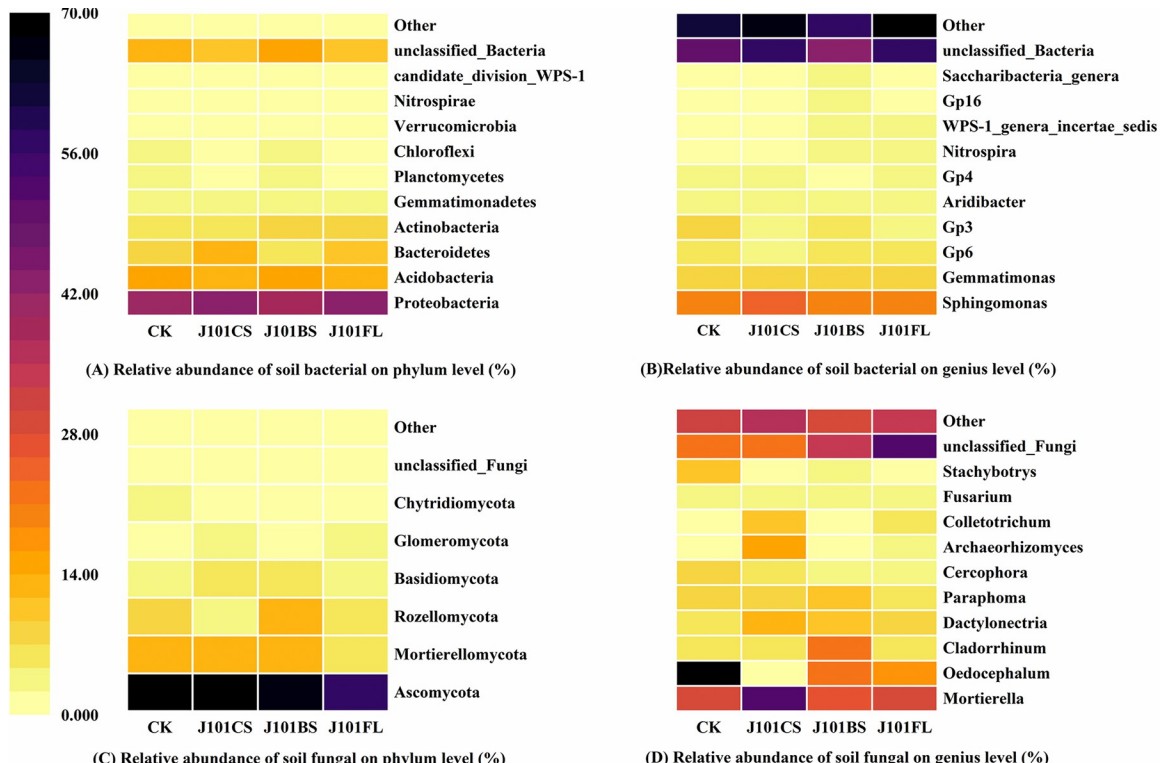

**Fig 4. The relative abundance of bacterial and fungal communities in maize rhizosphere soil.** (A) Changes in the relative abundance of bacteria at the phylum level; (B) changes in the relative abundance of bacteria at the genius level; (C) changes in the relative abundance of fungi at the phylum level; and (D) changes in the relative abundance of fungi at the genius level.

3.0. Therefore, a redundancy analysis (RDA) was used to analyze the relationship between the maize growth indicators and the soil properties. The RDA revealed that the contribution of the first axis to maize growth was 85.16%, and the total contribution for maize growth reached 97.66%. As shown in Fig 5, the height, stem thickness, dry weight, and SOD and POD activity in the shoot biomass and the roots of maize were positively associated with the residual fraction of Pb in the soil. Meanwhile, the maize's height, stem thickness, and fresh weight and the SOD and POD activity in the roots of maize were also positively associated with the sub-stable fractions of the soil Pb, such as the strong organic-bound, iron-manganese-oxide-bound, and humic-acid-bound fractions of Pb. However, the content of MDA and Pb in the maize plants and the SOD and POD activity in the leaves of maize had a positive association with the ion-exchange fractions of Pb in the soil. There was a positive association between the soil ACP and the soil AP content. The content of AP in the soil was positively associated with both the P content in the roots of maize and the nitrogen content in the shoot biomass and the roots of maize.

## Discussion

### Mechanism of P solubilization by phosphate-solubilizing bacteria

Previous studies have shown that PSB can significantly increase the soluble P in soil [42, 43]. In this study, the Pb-resistant PSB screened from tailings were identified as *P. rwandensis* using a phylogenetic tree comparison (Fig 1D). The PSB *P. rwandensis* not only dissolved $Ca_3(PO_4)_2$, but also demonstrated a tolerance to Pb (Fig 1A and 1B). We found that the

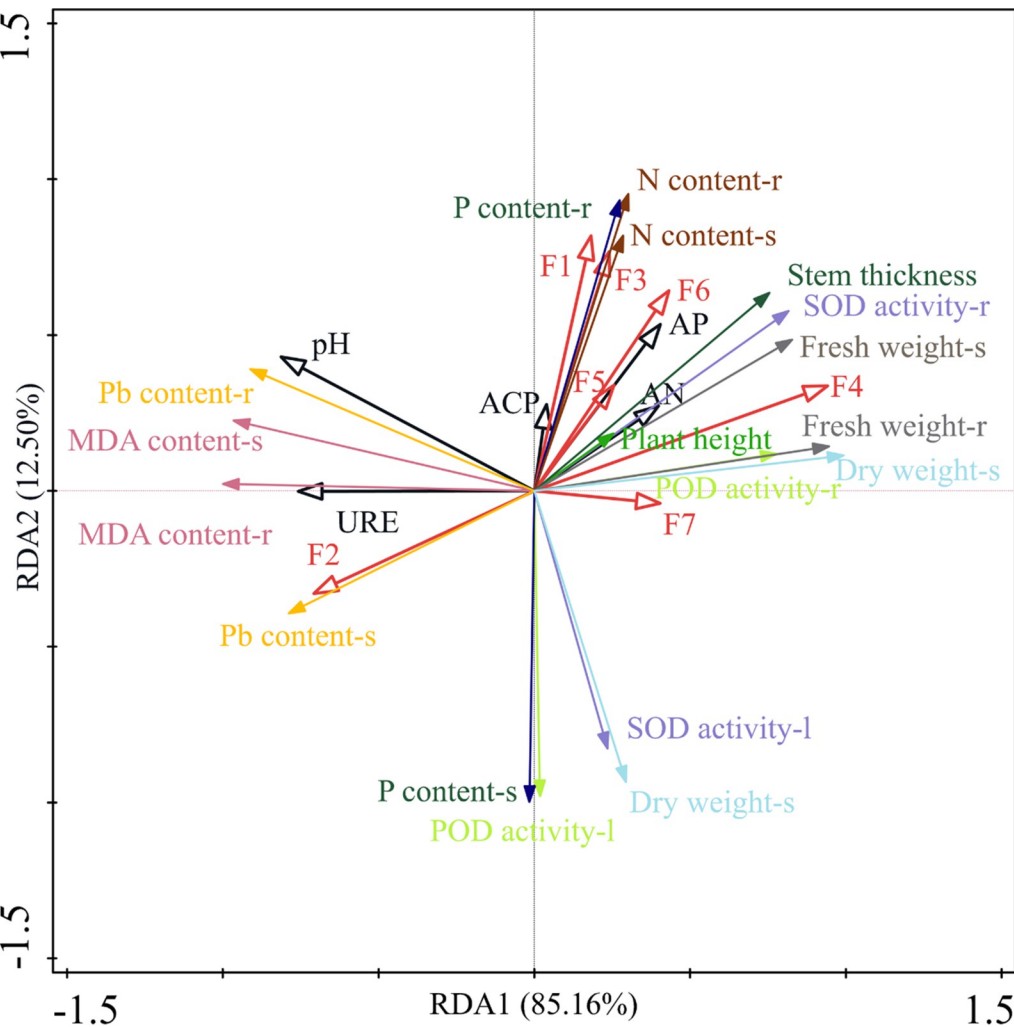

**Fig 5. RDA of maize agronomic traits and biomass in relation to soil physicochemical properties and different forms of Pb. Note:** "-s" represents the shoots of maize, "-l" represents maize leaves, and "-r" represents maize roots; F1: water-soluble forms of Pb; F2: ion-exchange forms of Pb; F3: carbonate-bound forms of Pb; F4: humic-acid-bound forms of Pb; F5: iron-manganese-oxide-bound forms of Pb; F6: strong organic forms of Pb; F7: residual forms of Pb; AP: available phosphorus; AN: alkaline nitrogen; ACP: acid phosphatase activity; and URE: urease activity.

organic acids produced by the PSB *P. rwandensis* were responsible for an important mechanism facilitating the solubilization of P from mineral phosphates through acidification and mental chelation. Generally, PSB release organic acid metabolites during biochemical processes, and they can reduce the environmental pH [44–47]. These organic acids contain multiple carboxyl or hydroxyl groups, which can chelate bound $Ca^{2+}$ cations and convert phosphate ions into soluble phosphates, resulting in the release of the bioavailable P [48–50]. The solubilization capacity of $Ca_3(PO_4)_2$ by bacteria isolated from mineral phosphates has been found to correlate with the organic acids produced by cowpea nodule strains [51], which aligns with our observations. Contrarily, some studies have shown that inoculating soil with the PSB *Bacillus thuringiensis* and *P. ananatis* can lead to an increase in the pH within the lettuce rhizosphere. This pH change is likely due to alterations in the soil microbial community and a higher release of $HCO_3^-$ (or $OH^-$) compared to the $H^+$ released by plant roots, as the result of a more significant uptake of anions ($PO_4^{3-}$) over cations [52]. However, the tolerance of *P.*

*rwandensis* to $Pb^{2+}$ may be due to the fact that *Pantoea* may detoxify the heavy metals Pb, cadmium (Cd), Zn, and mercury (Hg) by regulating the ATPases involved in metal transport and the copper-transporting P-type ATPase gene copA [25].

## Impacts on growth of maize

Inoculated PSB may improve plant development and protect plants against heavy metals through various mechanisms, such as phosphate solubilization, IAA, endogenous oxidants, and the immobilization of heavy metals [25]. In heavy-metal-contaminated soil, plants may improve endogenous oxidants (ROS) to alleviate the stress of free radicals inside the cells, which are produced under heavy metal stress [53]. In this study, the inoculated *P. rwandensis* significantly increased the activities of SOD and POD in maize roots (Fig 2B and 2C). This indicates that an increase in the activity of SOD and POD enhances the removal of superoxide anion radicals ($O^{2-}$) from maize roots, and peroxidase further detoxifies them. This is consistent with the results of Ali [53] and Wang [54], who demonstrated that inoculated *Enterobacter bugandensis*, *Serratia marcescens*, and *Bacillus anthracis* in Cd, Pb, and chromium (Cr)-contaminated soil can increase the SOD and POD of *Sesbania sesban* and *Ipomoea aquatica* Forssk. Reactive oxygen species (ROS) are produced more frequently in plant cells as the heavy metal content increases in the plant, which damages cellular constituents. The ROS accumulation further stimulates the production of MDA within the cells; therefore, the MDA content in cells is significantly positively correlated with the heavy metal accumulation in plants [55, 56]. The soil inoculation experiment showed that the Pb accumulation in the maize biomass was reduced by the inoculation of *P. rwandensis* (Fig 2A), which may have resulted in a decrease in the MDA in maize.

An inoculation with PSB may promote plant development and nutrient uptake by improving soil properties. Microorganisms producing IAA can stimulate cell elongation or affect cell division to increase root growth and lengthen the roots in plants, which leads to a larger root surface area and allows plants to obtain more nutrients from the soil [57]. The genus *Pantoea* has the genes ipdC and amiE, which code for the production of IAA, and these genes have two operons (speAB and speDE) that have been linked to the formation of lateral roots and the production of spermidine to relieve oxidative stress [58].

These experimental results indicate that the combined application of culture filtrate and bacterial isolates to maize roots is more effective at enhancing the activities of SOD and POD, as well as significantly reducing the MDA content, compared to either treatment alone. This suggests that microbes or their metabolic products can activate internal defense mechanisms in plants. When used together, they may act synergistically through different pathways, resulting in a more sensitive response of maize roots to environmental stress and initiating multiple signaling pathways to strengthen stress resistance [59]. During the combined application, the specific modulation of the signal molecule levels could positively affect the increase in the SOD and POD activities while concurrently decreasing the MDA content. Since these microbial inoculants are primarily applied at the root level, where they first interact with the root system, they likely alter the rhizosphere environment predominantly [60]. Additionally, because it takes longer for signal transduction or the transport of metabolic products to reach the leaves, the direct effects on the leaf antioxidant systems might not be as pronounced as those on the roots. Moreover, their levels can dynamically change depending on the environmental stresses, growth cycles, etc., making significant changes difficult to observe if the peak values are not captured during testing. Furthermore, when facing challenges such as poor soil conditions, plants may prioritize resource allocation towards tissues that require protection or functional enhancement most urgently to ensure absorption efficiency and survival rates.

An inoculation with *P. rwandensis* significantly increased the soil-available P content and increased maize's N uptake (Fig 3A and 3B). A decreased soil pH dissolved the insoluble inorganic P and then increased the amount of bioavailable P in the soil [54]. Meanwhile, the inoculation of PSB increased the soil ACP activity (Table 3). The ACP hydrolyzed phospho-diester and anhydrides in the soil to release inorganic P, which resulted in an increase in the bioavailable P in the soil [61–63]. In this study, an inoculation with *P. rwandensis* led to an increase in the AP nutrients in the soil. P is one of the most fundamental nutrients for plants; it is extensively involved in many key physiological and metabolic processes within plant organisms, including cell division and development, energy transport, signal trans-duction, the biosynthesis of macromolecules, photosynthesis, and respiration [64]. Indeed, it was found that there was a significant increase in N uptake by maize, although there was no significant change in the available N content in the soil (Fig 3A). The improved P supply due to the PSB inoculation may regulate ATP hydrolysis to release energy, which would affect metabolic processes such as protein and amino acid synthesis, so that the N content of the plant increases [65].

It has been reported that Pb in soil is converted into residual Pb fractions, such as Pb phosphate compounds, and its phytoavailability is significantly reduced [8]. We found that the residual Pb fractions in soil were significantly increased by an inoculation with *P. rwandensis*. These results confirm that the inoculation of PSB *P. rwandensis* promoted Pb immo-bilization, which may have reduced the accumulation of Pb in the maize biomass. This is consistent with the work by Liu, who demonstrated that *Pantoe sp*. PSB effectively immobi-lized Pb in lettuce rhizosphere soil [52]. Because of the P-releasing activity of PSB, the increase in solubilizing P from mineral phosphates could facilitate the formation of residual Pb phosphorate compounds in the soil. The plant height, stem thickness, and biomass of maize were positively associated with the residual Pb fraction in the soil (Fig 5). These find-ings align with the prevalent view that an increase in soil AP promotes the formation of insoluble Pb compounds. However, further research is needed to elucidate the process underlying the formation of Pb phosphate compound fractions. However, it was puzzling to us that the combined application of culture filtrate and bacterial isolates had a lesser effect on increasing the ACP, AP, and residual Pb content than the treatments with culture filtrate or the bacterial isolates inoculated individually. The relationships between the two types of inocula and the soil are complex, and the elucidation of the mechanisms requires further in-depth investigation.

In our research, we observed an interesting phenomenon: although the use of either cul-ture filtrate or bacterial isolates alone promoted the growth of maize, their combined appli-cation did not show a significant improvement. Consequently, we hypothesize that this phenomenon may have resulted from certain metabolic products (such as antibiotics, acids, etc.) or signaling molecules in the culture filtrate interacting with the added live bacteria. Such interactions might have hindered the effective colonization of the bacteria in the plant rhizosphere and affected their ability to enhance plant-growth-promoting functions. Addi-tionally, it should be noted that the accumulation of certain secondary metabolites or endo-toxin components can reach concentrations that have mildly toxic effects on maize seedlings when applied concurrently. Consequently, this may have attenuated the growth-enhancing impact on maize. It should be noted that not all benefits can be achieved simply by combining two treatment methods. Correct proportions and precise control over timing are crucial. Attention should also be paid to the possibility that large amounts of metabolic waste in the culture filtrate could have altered the soil pH values, thereby affecting nutrient absorption.

## Impact on soil microbial community diversity

Microbial distributions can be markedly changed by inoculating functional microbes, but the trend in the disturbance of the microbial community was inconclusive [66, 67]. Our results show that the inoculation of functional PSB promoted the biodiversity of the fungal community, affecting this community more significantly than the bacterial communities. This phenomenon may have been related to the competition for resources between PSB and soil microbial groups, making it more competitive by providing attachment points and inhibiting bacterial growth. In addition, the inoculated *P. rwandensis* was resistant to Pb (Fig 1B), so its inoculation likely reduced the abundance of the dominant phylum *Ascomycota* in the fungal community and increased the dominance of the sub-dominant genus *Oedocephalum*, which improved the biodiversity of the fungal community by reducing the competitiveness of the dominant phylum and genera of fungi and providing resources for the propagation of other fungi.

The application of the culture filtrate with the secretions of *P. rwandensis* and inoculated *P. rwandensis* increased the relative abundance of the dominant *Proteobacteria* in the bacterial community, and the secretions of *P. rwandensis* promoted a higher abundance of the genus *Sphingomonas*. This result was consistent with the studies in [68], in which an inoculation of *Pseudoalteromonas agarivorans* promoted a higher abundance of the genera *Proteobacteria* and *Sphingomonas* in the rhizosphere soil of *Brassica chinensis* L. in Pb-contaminated soil. The genus *Proteobacteria* enables the fixation of heavy metals and promotes plant growth, and the genus *Sphingomonas* has a high tolerance to heavy metals through absorption, precipitation, and oxidation–reduction. These microorganisms are the dominant genera responsible for metal immobilization and plant growth promotion [68]. In this study, the increase in the abundance of the *Proteobacteria* and *Sphingomonas* genera indicated that the inoculation of PSB promoted the relative abundance of microorganisms with a resistance to the heavy metal Pb. However, the dominance of the fungus *Mortierella* was increased after the application of culture filtrate with the secretions of *P. rwandensis* (Fig 4D). Some studies have shown that the fungus *Mortierella* has a high tolerance and adsorption capacity for heavy metals and can effectively reduce the bioavailability of Pb [69, 70].

In this study, neither the culture filtrate, the bacterial isolates, nor their combined application had a significant impact on the diversity of the soil bacterial community. This may have been due to differences in resource utilization between the fungi and bacteria, which could have provided more or novel resources (such as secondary metabolites produced by certain bacteria) for the fungi. Certain extracellular enzymes or active molecules, such as antibiotics, that might have been present in the culture filtrate could have inhibited specific types of bacteria or altered their genomic expression patterns [71]. If these active molecules suppressed some dominant bacteria, they may have freed up new niches for other non-dominant fungi. Additionally, the culture filtrate and the bacterial isolates could have led to changes in the local soil structure and the pH value, both of which are important factors influencing the micro-ecological balance. Generally speaking, fungi are better adapted to acidic environments and can survive under compacted or anoxic conditions [72].

## Conclusion

The current study demonstrated the following: (i) The Pb-resistant bacterium *P. rwandensis* facilitates the solubilization of insoluble calcium phosphate ($Ca_3(PO_4)_2$), exhibiting a notable tolerance to $Pb^{2+}$. Its primary mechanism for mineral P solubilization is through a pH reduction, achieved by secreting various organic acids. (ii) Inoculations with *P. rwandensis* and its metabolites significantly increased the residual fraction of Pb in the soil, suggesting that the

Pb-resistant PSB induced the formation of fewer mobile Pb fractions in the soil. Furthermore, an inoculation with *P. rwandensis* and its metabolites enhanced the soil fungal biodiversity and markedly increased the abundance of heavy-metal-tolerant microorganisms such as *Proteobacteria*, *Mortierella*, and *Sphingomonas* in the soil. (iii) An inoculation with *P. rwandensis* and its metabolites promoted robust growth in maize plants while substantially reducing the Pb accumulation within the plant biomass and impeding the translocation of Pb from the roots to the shoots. Additionally, *P. rwandensis* effectively boosted the antioxidant enzyme activities in maize roots and lowered the MDA levels in the plants, thereby enhancing their resistance to heavy metal toxicity from Pb. In summary, this research investigated how *P. rwandensis* affects the immobilization of the heavy metal Pb in contaminated soil while also making preliminary assessments regarding the factors influencing maize growth under these conditions. The indigenous bacteria *P. rwandensis* demonstrated potential capabilities for immobilizing the heavy metal Pb within soils, which holds promise for remediating environments burdened by Pb contamination. However, it is important to note that further studies are needed to fully elucidate the mechanisms behind the species transformations of Pb within different soil systems.

## Supporting information

**S1 Fig. Analysis of dissolved P, Pb tolerance, and secreted metabolites of J101.** (A) The change in the dissolved P content and the pH value in the broth medium; (B) the OD values at 600 nm after culturing the PSB in fermentation liquid with $Pb^{2+}$ concentrations of 0, 250, 500, 750, or 1000 mg·L-1 for 72 h; (C) the content of organic acid and IAA secreted by the PSB; (D) the phylogenetic tree of J101 based on the 16S rDNA gene sequences; and (E) the root systems of maize under different treatments.
(DOCX)

**S2 Fig. The effects of different treatments on Pb uptake, antioxidant enzyme activity, and malondialdehyde content in maize.** (A) Pb absorption by maize; (B) changes in superoxide dismutase (SOD) activity in maize; (C) changes in peroxidase (POD) activity in maize; and (D) changes in malondialdehyde (MDA) content in maize. **Note:** Different lowercase letters indicate significant differences between groups.
(DOCX)

**S3 Fig. Changes in the uptake of the nutrient elements N and P by maize.** (A) N uptake by maize; (B) P uptake by maize. **Note:** Different lowercase letters indicate significant differences between groups.
(DOCX)

**S4 Fig. The relative abundance of bacterial and fungal communities in maize rhizosphere soil.** (A) Changes in the relative abundance of bacteria at the phylum level; (B) changes in the relative abundance of bacteria at the genius level; (C) changes in the relative abundance of fungi at the phylum level; and (D) changes in the relative abundance of fungi at the genius level.
(DOCX)

**S5 Fig. RDA of maize agronomic traits and biomass in relation to soil physicochemical properties and different forms of Pb. Note:** "-s" represents the shoots of maize, "-l" represents maize leaves, and "-r" represents maize roots; F1: water-soluble forms of Pb; F2: ion-exchange forms of Pb; F3: carbonate-bound forms of Pb; F4: humic-acid-bound forms of Pb; F5: iron-manganese-oxide-bound forms of Pb; F6: strong organic forms of Pb; F7: residual

forms of Pb; AP: available phosphorus; AN: alkaline nitrogen; ACP: acid phosphatase activity; and URE: urease activity.
(DOCX)

**S1 Table. Effects of different treatments on agronomic traits and biomass of maize.** Note: Different lowercase letters in the table column-wise indicate significant differences between groups at $p < 0.05$.
(DOCX)

**S2 Table. Effects of different treatments on Pb forms in soil. Note:** Different lowercase letters in the table row-wise indicate significant differences between groups at $p < 0.05$.
(DOCX)

**S3 Table. Effects of different inoculation on soil properties. Note:** Different lowercase letters in the table column-wise indicate significant differences between groups at $p < 0.05$. AP: available phosphorus; AN: alkaline nitrogen; ACP: acid phosphatase activity; and URE: urease activity.
(DOCX)

**S4 Table. The α-diversity index of bacterial and fungal communities in maize rhizosphere soil. Note:** Different lowercase letters in the table column-wise indicate significant differences between groups at $p < 0.05$.
(DOCX)

**S1 Dataset. Raw data for figures and tables in the paper.**
(XLSX)

## Acknowledgments

This project was supported by Yunnan Collaborative Innovation Center for Plateau Lake Ecology and Environmental Health. We also acknowledge the support of the College of Agronomy and Life Sciences, Kunming University for providing the greenhouse and laboratory facility for experimenting. We appreciated professor Xuexiu Chang for revising the grammar of the paper.

## Author Contributions

**Conceptualization:** Chengqiang Zhu.

**Data curation:** Runhai Jiang.

**Funding acquisition:** Xiuli Hou.

**Investigation:** Mei Zhang.

**Methodology:** Chengqiang Zhu, Shaofu Wen, Yani Zhang.

**Project administration:** Xiuli Hou.

**Software:** Runhai Jiang.

**Visualization:** Mei Zhang.

**Writing – original draft:** Runhai Jiang.

**Writing – review & editing:** Xiuli Hou.

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
