## [Decision Letter · Decision Letter 0]

6 Feb 2024

PONE-D-23-42620Pb-resistant phosphate-solubilizing bacteria promote maize’s growth by altering Pb accumulation in biomass and soil Pb immobilizationPLOS ONE

Dear Dr. Hou,

Thank you for submitting your manuscript to PLOS ONE. After careful consideration, we feel that it has merit but does not fully meet PLOS ONE’s publication criteria as it currently stands. Therefore, we invite you to submit a revised version of the manuscript that addresses the points raised during the review process.

We look forward to receiving your revised manuscript.

Kind regards,

Durgesh Kumar Jaiswal, Ph.D.

Academic Editor

PLOS ONE

“This work is funded by National Natural Science Foundation of China (42167009), Joint Fund Project of Yunnan Provincial Universities (2018FH001-004), Scientific Research Fund Project of Yunnan Provincial Department of Education (2022Y712, 2023Y0874), The International Joint Innovation Team for Yunnan Plateau Lakes and Laurentian Great Lakes (funded by Yunnan Provincial Education Department), College Students' Innovation and Entrepreneurship project (202111393070).”

5. PLOS requires an ORCID iD for the corresponding author in Editorial Manager on papers submitted after December 6th, 2016. Please ensure that you have an ORCID iD and that it is validated in Editorial Manager. To do this, go to ‘Update my Information’ (in the upper left-hand corner of the main menu), and click on the Fetch/Validate link next to the ORCID field. This will take you to the ORCID site and allow you to create a new iD or authenticate a pre-existing iD in Editorial Manager. Please see the following video for instructions on linking an ORCID iD to your Editorial Manager account: https://www.youtube.com/watch?v=_xcclfuvtxQ.

6. Please amend the manuscript submission data (via Edit Submission) to include author Xuexiu Chang.

7. Please include your tables as part of your main manuscript and remove the individual files. Please note that supplementary tables (should remain/ be uploaded) as separate "supporting information" files.

8. We notice that your supplementary figures are uploaded with the file type 'Figure'. Please amend the file type to 'Supporting Information'. Please ensure that each Supporting Information file has a legend listed in the manuscript after the references list.

Additional Editor Comments:

We have thoroughly reviewed your manuscript by reviewers and editors, and it requires major revision for publication. Please follow the comments and resubmit your revised manuscript with the reviewers' responses.

Reviewers' comments:

Reviewer's Responses to Questions

**Comments to the Author**

1. Is the manuscript technically sound, and do the data support the conclusions?

Reviewer #1: Partly

Reviewer #2: Partly

Reviewer #3: Yes

2. Has the statistical analysis been performed appropriately and rigorously? 

Reviewer #1: No

Reviewer #2: Yes

Reviewer #3: Yes

3. Have the authors made all data underlying the findings in their manuscript fully available?

Reviewer #1: No

Reviewer #2: Yes

Reviewer #3: Yes

4. Is the manuscript presented in an intelligible fashion and written in standard English?

Reviewer #1: No

Reviewer #2: No

Reviewer #3: Yes

5. Review Comments to the Author

Reviewer #1: This study examines the efficacy of Pantoea rwandensis in reducing the availability and plant uptake of lead (Pb) in Pb-contaminated soil. The results of the study indicate that P. rwandensis solubilized tricalcium phosphate by secreting various organic acids and also improved maize growth and enhanced the diversity of fungi and bacteria in Pb-contaminated soils. From the results, it was concluded that the P-solubilizing P. rwandensis affected the transformation of Pb fractions in the soil through alterations in the functions of the soil bacteria thereby decreasing Pb accumulation in maize and improving its growth. Although the results are interesting, there are several concerns as mentioned below that need attention. As the changes are numerous to list here, I marked them directly in the annotated manuscript.

1. The manuscript should be language edited to improve readability. The sentences are wordy and incomplete making reading and understanding difficult. I would suggest the authors seek the help of a colleague with good proficiency in English to language edit the manuscript.

2. There are already studies where species in Pantoea are reported to possess nutrient solubilizing, plant growth, and heavy metal tolerance (e.g., https://doi.org/10.1111/1462-2920.14946; https://doi.org/10.3390/microorganisms8020153; https://doi.org/10.1007/s11274-019-2744-4; https://doi.org/10.3390/microorganisms9081661). Unfortunately, previous studies available on the plant-growth-promoting activities of Pantoea are not adequately summarized in the introduction.

3. The study is confounded by methodological shortfalls. For instance, the phosphatic source used for the determination of phosphate solubilization is not correct. Tricalcium phosphate is a weak phosphatic source that can be solubilized by any acid-producing bacteria including Escherichia coli. To determine true phosphate-solubilization, a harder-to-solubilize phosphatic source like aluminium phosphate or iron phosphate has to be used. If the bacterial isolate can solubilize these hard-to-solubilize phosphatic sources then it will invariably solubilize tricalcium phosphate. For more details, see https://doi.org/10.1007/s00374-012-0756-4.

4. Similarly, Salkowski’s reagent can react with a wide range of indole compounds (e.g., indole-acetamide, indole-3-pyruvic acid) to produce colour and is not specific to indole acetic acid. To be more precise, a more reliable technique using HPLC or LC-MS should be used for determining indole acetic acid production (consult https://doi.org/10.3389/fchem.2018.00265; https://doi.org/10.1128/aem.61.2.793-796.1995).

5. The methods section requires a rewriting by incorporating missing information. Important information like the plant species from which the soil samples were collected, tailing soil characteristics, the depth and quantity of soil collected, source of maize seeds, experimental design (completely randomized block design, etc.), etc., are missing.

6. Was any standard method used for collecting the rhizosphere soil? If not state how rhizosphere soil was differentiated from the surrounding bulk soil.

7. Explain how different concentrations of Pb were fixed to test the tolerance of the bacterium to the heavy metal.

8. The criteria used for the selection of the isolate J101 for further studies are missing.

9. Parametric analysis like ANOVA requires normal distribution of data. Therefore it is important to check the homogeneity of the data. If the data fails to satisfy normality, then it should be transformed. Non-parametric analysis should be used if the data fails to satisfy normality even after transformation. Moreover, although Post Hoc analysis (Duncan’s multiple range test) was used mean separations are not presented for those in Figure 1.

10. The total number of PSB isolates in the tailing soil samples is missing in the results. Remove the data that are already there in the tables and figures from the results.

11. As the treatments are compared with a control Dunnett Post Hoc analysis would be more appropriate than DMRT.

12. Many parts of the discussion are repetitions of the results. Some of the results are not adequately discussed. For instance, the combined application of culture filtrate and the bacterial isolate failed to improve maize growth compared to their inoculation. Nevertheless, the reasons for this decreased intensity in plant growth are not adequately discussed.

Other comments:-

13. Lines 44–45: Avoid using words that are already there in the title as keywords.

Reviewer #2: The manuscript entitled “Pb-resistant phosphate-solubilizing bacteria promote maize’s growth by altering Pb accumulation in biomass and soil Pb immobilization”. (PONE-D-23-42620)

The authors contribute to evaluate that application of phosphate solubilizing bacteria (PSB) to reduce the phytoavailability of Pb uptake by maize plant. The main results were inoculation with P. rwandensis could promote the transformation of Pb fractions, and affected the soil bacteria community, and finally promoted the growth of maize. The work is helpful in usage of PSB strain. However, there still several major issues should be revised. Moreover, the language should be further improved.

Major concerns:

The goal of this work should be clearer. I think more information should be summarized to present the novelty of this work. What is the great difference of this strain from previous studies, and the new finding in this work should be provided. Why is the maize plant used in this study? The reasons should be mentioned in the Introduction section.

Minor concerns:

The authors should revise the text. Lots of minor revisions should be paid more attention.

For instance, Line 64 PSB when the first time to describe, it should be give full name.

Line 68 to 70, should be revised.

Line 89 to 90, should be revised.

Line 98 to 100, should be revised.

Line 153 strain J101

Line 168 should be revised.

Line 189 to 191 The software version and full website should be revised.

Line 208 and 212 please check and revise.

Line 227-232 This section should be revised, what is your main results.

Line 247 to 248 MDA please use full name in figure caption.

Line 369 please revise.

Line 378-379 please revise.

Line 393 please revise. Please use the author names, not reference number.

Line 401 please revise.

Line 439 please revise.

Line 469, 473, 475, 477, 479, this section should be completely revised.

Figure 1 please revise.

Figure 1 and Figure S1 and S2 can be merged into one figure.

Although I can not provide each minor revision, the authors should pay more attention to your text, especially the language. The goal is to make your work readable and useful to readers.

Reviewer #3: This is an interesting, clear, concise, and well-written manuscript. The introduction is relevant and theory based. Sufficient information about the previous study findings is presented for readers to follow the present study rationale and procedures. The methods are generally appropriate. I enjoyed the manuscript and believe that it will be a relevant contribution to the field. It only needs to undergo a few minor changes before its publication. This paper has a potential to be accepted, but some important points have to be clarified basically in tables and figures. Below I have provided numerous remarks to correct the text.

• In several instances the authors develop a unique theoretical framework, and I believe that they should highlight their originality much more.

• The manuscript contains an elaborate literature, but definitions of the key concepts are needed in the introduction.

• The authors draw on extensive empirical evidence. I believe that they can put forward their arguments much more confidently.

• Throughout the manuscript, there are several minor language mistakes. Therefore, I recommend a certified round of language editing before the paper is published.

• It is very difficult to understand the figures and tables without legend. The authors are requested to put the proper titles along-with figures and tables. The information given under the title of supporting information (line no 672-674) is not adequate, in which only information is given regarding S1 only not for S2 and it is also not very clear.

• The titles given in every table (for example in table no 1, Change of biomass in maize) is not adequate and not given proper message

• In table 3, authors should write the abbreviations URE, ACP AP and AN

• No doubt that figure are impressive but authors should describe regarding alphabets like a,b,c etc. in the figure

• In figure 5, what is RDA1 and RDA 2

• Clear that what does Phylogenetic tree indicates, properly discuss in your text.

• Line No. 214 to 216; no need to write this is the fig 1 title. Title should be written with figures

• Line no 233 to 236, text should be written with table 1

• Check line 251-256 for title. Is it for fig 1 or 2.

• Line 264-269 write title with figure 3

• Line 286 to 290 write title with respective table

• Line no 292 activity of acid phosphatase is ok

• Line 301-305 write title with table

• Line 314-318 write title with table 4

• Line 334-336 write title with figure 4

• Line no 352-365 write text with fig 5 and try to reduce the text of title

• English Grammar Mistakes in Technical Manuals

1.

Evaluation of the purposive approach to statutory interpretation

6: Finally I recommend that this paper be accepted after minor revision.

2.

6. PLOS authors have the option to publish the peer review history of their article (what does this mean?). If published, this will include your full peer review and any attached files.

Reviewer #1: No

Reviewer #2: No

Reviewer #3: No

---

## [Author Response · Author response to Decision Letter 0]

28 Feb 2024

The document for the "respond to reviewers" has been uploaded in the system, please check it in the attachment.

---

## [Decision Letter · Decision Letter 1]

18 Jun 2024

Pb-resistant Pantoea rwandensis promotes maize's growth by altering Pb accumulation in biomass and soil Pb immobilization

PONE-D-23-42620R1

Dear Dr. Hou,

We’re pleased to inform you that your manuscript has been judged scientifically suitable for publication and will be formally accepted for publication once it meets all outstanding technical requirements.

Kind regards,

Ying Ma, Ph.D.

Academic Editor

PLOS ONE

Additional Editor Comments (optional):

Reviewers' comments:

Reviewer's Responses to Questions

**Comments to the Author**

1. If the authors have adequately addressed your comments raised in a previous round of review and you feel that this manuscript is now acceptable for publication, you may indicate that here to bypass the “Comments to the Author” section, enter your conflict of interest statement in the “Confidential to Editor” section, and submit your "Accept" recommendation.

Reviewer #1: All comments have been addressed

Reviewer #2: (No Response)

2. Is the manuscript technically sound, and do the data support the conclusions?

Reviewer #1: Yes

Reviewer #2: (No Response)

3. Has the statistical analysis been performed appropriately and rigorously? 

Reviewer #1: Yes

Reviewer #2: (No Response)

4. Have the authors made all data underlying the findings in their manuscript fully available?

Reviewer #1: Yes

Reviewer #2: (No Response)

5. Is the manuscript presented in an intelligible fashion and written in standard English?

Reviewer #1: Yes

Reviewer #2: (No Response)

6. Review Comments to the Author

Reviewer #1: In this revised manuscript, the authors have addressed all the suggestions and changes from the previous version. The manuscript has significantly improved.

Reviewer #2: The manuscript entitled “Pb-resistant Pantoea rwandensis promotes maize's growth by altering Pb accumulation in biomass and soil Pb immobilization”. The authors have made a lot of major revisions of this manuscript. I can see that the main goals are clear now.

7. PLOS authors have the option to publish the peer review history of their article (what does this mean?). If published, this will include your full peer review and any attached files.

Reviewer #1: No

Reviewer #2: No

---

## [Editor Report · Acceptance letter]

1 Aug 2024

PONE-D-23-42620R1 

PLOS ONE

Dear Dr. Hou, 

I'm pleased to inform you that your manuscript has been deemed suitable for publication in PLOS ONE. Congratulations! Your manuscript is now being handed over to our production team.

Kind regards, 

on behalf of

Dr. Ying Ma 

Academic Editor

PLOS ONE